

# Global ship accidents and ocean swell-related sea states

Zhiwei Zhang[1, 2], Xiao-Ming Li[2, 3]

[1] College of Geography and Environment, Shandong Normal University, Jinan, China
[2] Key Laboratory of Digital Earth Science, Institute of Remote Sensing and Digital Earth, Chinese Academy of Sciences,
Beijing, China
[3] Hainan Key Laboratory of Earth Observation, Sanya, China

*Correspondence to:* X.-M. Li (E-mail: lixm@radi.ac.cn)

**Abstract.** With the increased frequency of shipping activities, navigation safety has become a major concern, especially when
economic losses, human casualties and environmental issues are considered. As a contributing factor, sea state conditions play
a significant role in shipping safety. However, the types of dangerous sea states that trigger serious shipping accidents are not
well understood. To address this issue, we analyzed the sea state characteristics during ship accidents that occurred in poor
weather or heavy seas based on a ten-year ship accident dataset. The sea state parameters, including the significant wave height,
the mean wave period and the mean wave direction, obtained from numerical wave model data were analyzed for selected ship
accidents. The results indicated that complex sea states with the co-occurrence of wind sea and swell conditions represent
threats to sailing vessels, especially when these conditions include close wave periods and oblique wave directions.

## 1 Introduction

The shipping industry delivers 90% of all world trade (IMO, 2011). It is currently a thriving business that has experienced
increases in both the number and size of ships. However, due to the frequency of shipping activities, ship accidents have
become a growing concern, as have the following destructive consequences: casualties, economic losses and various types of
environmental pollution.

Investigations into the causes of shipping accidents show that over 30% of the accidents are caused by poor weather, and
an additional 25% remain completely unexplained (Faulkner, 2004). Due to these dangerous uncertainties, accidents that
involve poor weather and severe sea states should be further studied for shipping safety.

However, under changing weather conditions, the sea surface is too complex to predict, especially on a small timescale
(Kharif et al., 2009). The sea surface is composed of random waves of various heights, lengths and periods. Meanwhile,
different kinds of waves emerge frequently, among them, wind sea and swells are two main types of ocean waves classified by
wave generation mechanisms. Wind sea waves are directly generated by local winds, and when wind-generated waves
propagate without receiving further energy from wind, they gradually grow to a swell.

Meteorologists and oceanographers generally work with statistical parameters, such as the significant wave height ($H_s$),
wave period ($T$) and wave direction ($D$), to represent a given sea state. Additionally, the wave spectrum that gives the
distribution of wave energy among different wave frequencies ($f, f = 1/T$) is analyzed in some studies to better understand



wave dynamics. Note that a typical ocean wave spectrum with two peaks (e.g., one from the distance swell and the other generated by the local wind) will be much more complicated and variable.

Regarding sea state parameters, $H_s$ is usually a practical indicator of the sea state during marine activities. Indeed, some studies have shown that accident areas coincide with the zones of the highest $H_s$, e.g., in an analysis of ship accidents that
occurred in the North Atlantic region (Guedes et al., 2001). A high wave height is no doubt a threat for ships, yet some ships wreck at relatively low wave heights and high wave steepness sea states (Toffoli et al., 2005).

A sea state with a narrow wave spectrum was observed during several major ship accidents, including the 'Voyager' accident (Bertotti and Cavaleri, 2008), the Suwa-Maru incident (Tamura et al., 2009), the Louis Majesty accident (Cavaleri et al., 2012) and the Onomichi-Maru incident (In et al., 2009; Waseda et al., 2014). Studies have assumed that the narrowed wave
spectrum is primarily generated by the nonlinear coupling of swell and wind sea (or swell and swell) (Bertotti and Cavaleri, 2008; Tamura et al., 2009; Cavaleri et al., 2012; Waseda et al., 2012). During such wave couplings, the wave energy from one wave system (wind sea or swell) is enhanced and transformed to the other wave system (usually a swell) (Tamura et al., 2009; Waseda et al., 2014).As a result, the wave energy transformation produces a steep swell, with a large wave energy and extreme wave height (Bertotti and Cavaleri, 2008).

The oblique angle between two waves is another important condition involved in the interaction of wave systems. Various traveling angles have been discovered during ship accidents, ranging from 10° (Onorato et al., 2010) to 60° (Tamura et al., 2009). The features noted above individually or simultaneously emerge during ship accidents or freakish sea states when swells and wind seas co-occur. Indeed, the co-occurrence of wind seas and swells can lead to dangerous seas, as demonstrated by the parametric rolling that occurred for the German research vessel Polarstern (Bruns et al., 2011), although extreme wave
heights were not observed.

In previous studies of ship accidents, researchers focused on only one severe accident when discussing the sea state dynamics in detail or based their studies on ship accident data to perform statistical analyses of classical sea state parameters (e.g., $H_s$ and $T$). To thoroughly investigate sea state parameters, we collected information on a large number of ship accidents and created a database for analysis. Additionally, we discussed the parameters in both wind sea and swell conditions. Statistical
analyses were performed on data obtained from the International Maritime Organization (IMO). The data include ten years of ship accidents (2001 – 2010) and 755 cases caused by bad weather or heavy seas. Because swells with large wave energies can represent a threat to maritime activities, 58 cases in which swells were reported as an important factor in the ship accident were selected. The detailed information discussed above are presented in section 2. Following an overview of the ship accidents (section 3), an analysis of the swell-related sea state conditions for these ship accident cases is presented in section 4. In section
5, two cases are illustrated to demonstrate the dynamic processes that ensue when wind sea and swell conditions occur during ship accidents. Finally, a summary and discussion are provided (section 6).



## 2 Data and Methods

### 2.1 Ship Accident Database

A ten-year (2001–2010) ship accident dataset was gathered from the Marine Casualties and Incidents Reports issued by the IMO. The dataset includes 3648 ship accidents, and each accident in the report includes the occurrence information, such as the

accident time and coordinates, initial event, summary, casualty type, ship type, etc. Since the primary information used in this study includes the accident time and coordinates, after excluding the events that failed to record these details, 1561 cases with exact geographical locations remained in the dataset.

According to the description of initial events, which provides clues regarding the accident causes recorded in the reports, those 1561 valid cases cover different kinds of cases triggered by natural factors and human factors. Because we focus on the

events that occurred in natural weather-related conditions, cases with the descriptions such as fire or explosion, improper operations, and lost persons were eliminated from the 1561 cases, while cases with keywords such as strong wind/gale/cyclone or heavy seas/rough waves were kept. After filtering, 755 weather-related accidents were obtained for the further analysis. An overview of these 755 cases is presented in section 3.

Furthermore, this study focuses on the cases that occurred in swell-related sea states. After examining all the summaries of the 755 cases, we kept 58 cases with clear descriptions of swell movement during ship accidents for the analysis of swell-related sea states. The detailed analysis is presented in section 4.

### 2.2 Numerical Wave Model Data

The ERA-20C numerical wave model data were obtained from the European Center for Medium-Range Weather Forecasts (ECMWF). The ECMWF uses atmosphere, land, surface, and ocean wave models and data to reanalyze the weather conditions

during the last century. The ERA-20C products describe the spatio-temporal evolution of the atmosphere (on 91 vertical levels, between the surface and 0.01 hPa), the land-surface (in 4 soil layers), and ocean waves (for 25 frequencies and 12 directions). The accuracy is improved by validation with ERA-40 data and operational archive results. Compared to the ERA-Interim dataset, ERA-20C has longer reanalysis coverage (24 h) for single-point data (Poli et al., 2013). The Ocean Wave Daily data in the ERA-20C dataset are available from 1900–2010 every 3 hours at a grid size of 0.125°. The data provide 33 reanalyzed

ocean wave parameters, and separate entries are included for swell and wind sea conditions.

## 3 Overview of Ship Accidents

In the ship accident dataset, 755 weather-related cases were distinguished and expounded in section 2. Hereafter, we provide an overview on these 755 cases in terms of the initial events, ship types and spatial distribution. Five types of initial events (Figure 1(a)) were assumed as weather-related ship accidents. The results show that the stranding/grounding type of accident ranks

first in terms of frequency and accounts for 40.3% of all accident cases. In addition, different types of ships respond differently




when they encounter potentially dangerous sea conditions because of their unique structures and functions. Among the 755 cases, general cargo vessel types experienced the highest proportion of accidents (32.3%) in rough weather and severe sea states, and they are followed by bulkers and fishing vessels. Collectively, these data highlight the types of ships that may require more attention during shipping activities (Figure 1(b)).

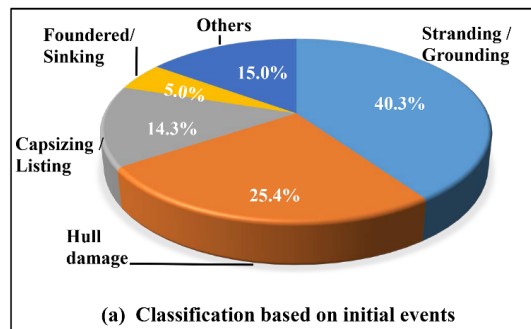
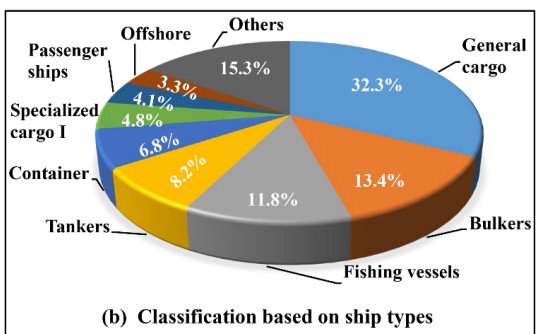

5   **Figure 1.** Classification of the 755 weather-related ship accidents based on initial events (a) and ship type (b). The accidents were recorded in the International Maritime Organization (IMO) database.

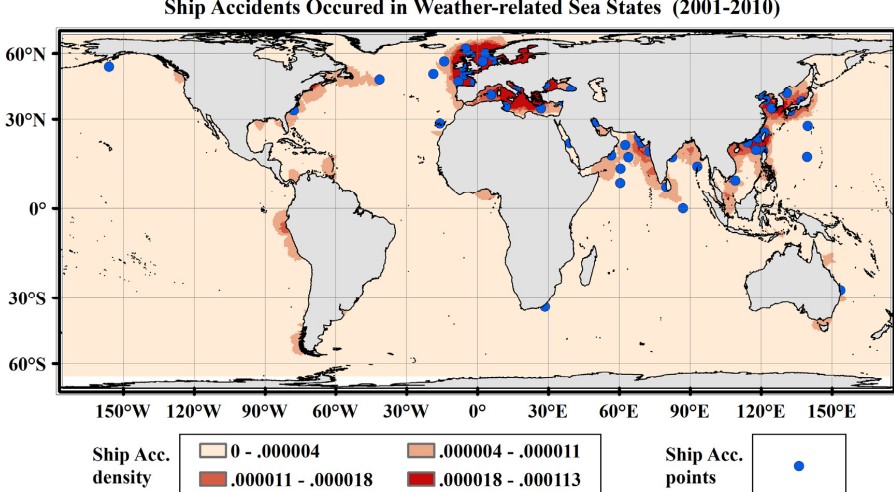

10  **Figure 2.** Geographical distribution of the ship accident density according to the 755 weather-related cases. The superimposed blue dots indicate the ship accidents (58 cases) that occurred in a swell sea state. The figure was created in ArcGIS for Desktop software (version 10.2.0.3348).

The spatial distribution of the 755 cases is presented in Figure 2. To construct the ship accident density graph, the research

15  area was divided into 188,325 raster cells with a cell size of 50 km×50 km. Then, a circle area with a radius of 500 km was




defined as the region around each cell center. The number of ship accidents that fell within each region was summed and divided by the area of the region, which provided the ship accident density. Additionally, 58 ship accidents that occurred in swell sea states have been superimposed as blue dots. The areas of deeper colors in the map reflect a higher density of ship accidents. Clearly, these accidents are densely distributed in the North Atlantic Ocean, the North Indian Ocean and the West

5 Pacific Ocean, which represent areas that coincide with the major shipping routes of Asian, European and North American countries.

Figure 2 shows that few accidents occurred in the open sea, although this result may have been related to the limited data recorded in the IMO database on severe open sea accidents that occurred from 2001–2010.

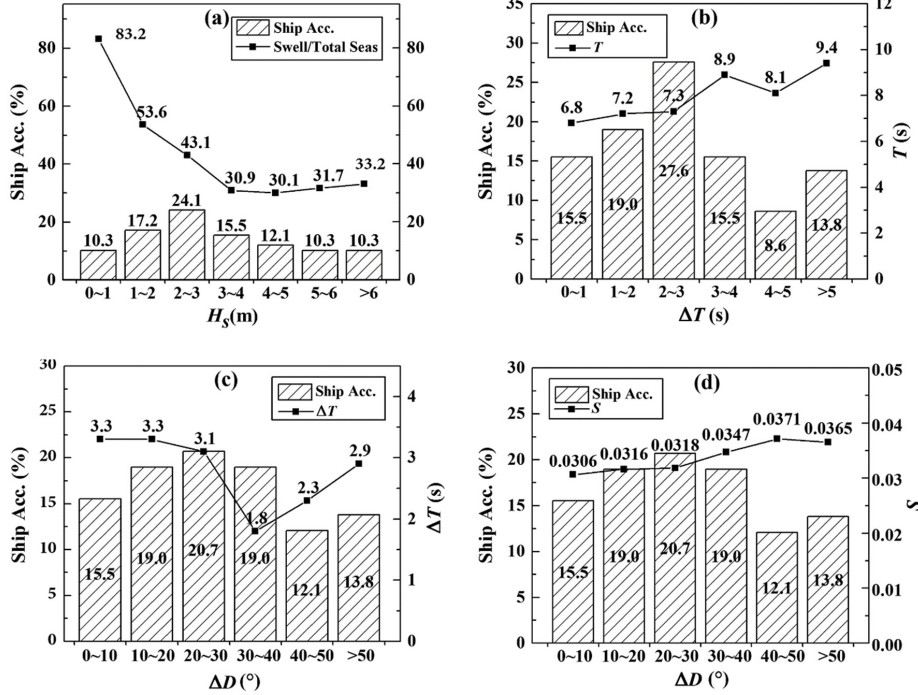

10 **Figure 3. Incidence rate of ship accidents: (a) at different significant wave heights ($H_s$, bar chart) and the proportional change in swell energy in the total sea (polygonal line); (b) with wave period differences ($\Delta T$, bar chart) and the mean wave period ($T$, polygonal line); (c) with wave direction differences ($\Delta D$) and $\Delta T$; and (d) with $\Delta D$ and the value of the wave steepness of the total sea ($S$)**





## 4 Analysis of the Sea State during Ship Accidents

As discussed in the Introduction section, the co-occurrence of wind sea and swell conditions is considered a potential causal factor that can lead to dangerous sea states for ships. In this section, we focus on 58 swell-related cases to discuss the sea state characteristics associated with wind sea and swell conditions. The sea states are described by three parameters: wave height, wave period and wave direction.

### 4.1 Wave Height

In terms of swell-related cases, both of the total sea wave height and swell wave height ($H_{sw}$) were analyzed. Moreover, the percentage of swell wave energy to the total sea energy was used in the analysis. Figure 3(a) shows the distribution of those values. The bar chart indicates that almost half of the cases occurred in an $H_s$ range of 0 m ~ 3 m, which is not enough to warrant a rough-sea warning. However, the proportion of the swell wave energy (i.e., the polygonal line in the graph) within this range is greater than 50%; thus, when ships sail in relatively low sea states, the increased contribution of the swell may lead to dangerous sea states that threaten shipping safety. Along with an increase in $H_s$, the proportion of swell wave energy to the total sea energy remains at approximately 30%, which reflects the increasing contribution of wind sea to worsening sea conditions when $H_s$ is greater than 3 m. In general, almost half of the swell-related cases occurred at $H_s$ values smaller than 3 m, which suggests that high wave height is not the only critical factor that triggers ship accidents. Indeed, other parameters may also play pivotal causal roles in these accidents. Therefore, additional wave parameters, including the wave period and wave direction, are subsequently examined for these accidents.

### 4.2 Mean Wave Period

Figure 3(b) depicts the relationship between the occurrence of specific ship accidents and wave period differences (bar charts) from swells and wind sea ($\Delta T$, i.e., the mean period of the swell minus the mean period of the wind sea). Furthermore, the mean wave period of the total sea ($T$, solid line) is also plotted in the graph. Approximately two-thirds of the cases occurred in sea states where $\Delta T$ was less than 3 seconds (s) and the value of $T$ approached 7 s, which represents a close wave period for swell and wind sea conditions in most cases. In other cases, the value of $T$ was larger than 8 s when $\Delta T$ was larger than 3 s. Overall, an upward trend can be observed with an increase in $\Delta T$, except for a slight fluctuation between 4 s and 5 s. Overall, a close mean wave period ($\Delta T$ <3 s) between swell and wind sea in a co-occurring sea state is an important factor for shipping accidents.

### 4.3 Mean Wave Direction

As noted in the Introduction, previous theoretical studies and ship accident analyses have indicated that crossing sea states (particularly crossing swell and wind sea states) may induce high waves and generate dangerous sea conditions. To investigate this issue further, the mean wave direction differences between the swell and wind sea ($\Delta D$, i.e., the absolute value of the mean





swell direction minus the mean wind sea direction) for all the swell-related ship accidents have been analyzed. Approximately half of the cases (55%) exhibited $\Delta D$ values less than 30° (Figure 3(c)), and the values of $\Delta T$ (indicated by the solid line superimposed on the bars) within this range were approximately 3 s before decreasing to 1.8 s at $\Delta D$ values ranging from 30° to 40°. During swell and wind sea interactions, the rate of change in swell energy under the influence of wind sea energy (Tamura

et al., 2009) reaches a maximum at approximately 40°(Masson, 1993). The $\Delta D$ range of 30° to 40° for the lowest value of $\Delta T$ (1.8 s) demonstrates the strong coupling between two waves. However, 45% of the accidents were associated with $\Delta D$ values larger than 30°. An angle of 30° appears to be a critical point for ship accidents because a rising trend in the $\Delta T$ line begins at this point. As the angle increases, the $\Delta T$ values decrease to below 3 s, and the sea state can be more easily transformed into a crossing sea state (Li, 2016; Onorato et al., 2010), which could pose a risk for ships.

Figure 3(d) is identical to Figure 3(c) except that the wave steepness of the total sea ($S$) has been added to the bar chart. In the present study, the wave steepness of the total sea is calculated via $S = 2\pi H_s/gT^2$. Along with an increase in $\Delta D$, a rising trend in wave steepness can be observed, although a slight fluctuation appears from 40°~50°. Wave steepness appears to be positively correlated with $\Delta D$, particularly when $\Delta D$ is approximately 50°. This is associated with a crossing sea state with a close wave period. Overall, large direction angle and wave steepness values appear to generate dangerous sea state conditions.

**5 Sea States of Typical Cases**

Based on the statistical analysis of the sea state characteristics presented above, we preliminarily conclude that close wave periods and oblique angles between co-occurring wind sea and swell conditions play important causal roles in ship accidents. In this section, two cases are presented to reveal the dynamic processes underlying co-occurring wind sea and swell conditions during ship accidents. One case occurred under a relatively low sea state, while the other case occurred at a high sea state.

Detailed sea state conditions for the two cases are presented in Figures 4 and 5. The first column in each figure indicates the wave model results 3 hours before the accident, whereas the second column indicates the results at the time closest to the occurrence of the accident.

The first accident occurred at approximately 18:45 on 24 September 2008 (UTC). The site (21°46.2′ N, 114°12.9′ E) was 25 nautical miles (nm) south of Hong Kong, China. After experiencing heavy winds and swell from a southeasterly direction,

the container ship Chicago Express, with a gross tonnage of 93,811, was suddenly hit by a particularly violent wave from starboard that caused casualties and damages.

At 18:00 UTC, the closest time to the accident time, the modeled $H_s$, $H_{ws}$ (wind sea wave height), and $H_{sw}$ values were 3.02 m, 2.45 m and 1.69 m, respectively, whereas at 3 hours before the accident (15:00 UTC), they were 2.82 m, 2.12 m and 1.74 m, respectively. Clearly, the values of $H_s$ and $H_{ws}$ increased, whereas $H_{sw}$ decreased. During the three-hour period, $H_s$

waves (4-meter contour line) moved toward the accident point. The contour line of high $T_{sw}$ (9 seconds) approached the accident area and transformed to a sharp angle shape. The wind sea period increased slightly from 6.0 s to 6.1 s due to the strong wind (13.7 m/s) and approached the swell wave period. Thereby, the 2-s contour line of $\Delta T$ moved to the accident point




during the 3-hour period, indicating a closer wave period process between the swell and wind sea in the ship accident area. In terms of wave directions, the wind sea (the light grey arrows) spread northward while the swells (the black arrows) spread eastward in the southern area and northward in the northern area. As a result, two distinct areas can be observed in the $\Delta D$ graphs: a small $\Delta D$ area in the northerly direction and a large $\Delta D$ area in the southerly direction. The boundary of the two

distinct areas (in the background color of deep blue) exhibited $\Delta D$ values of $40° \sim 60°$ and increased closer to the accident point. Meanwhile, $\Delta D$ at the accident site increased from $40.4°$ to $51.6°$. This angle interval is favorable for wave energy transformation between two nonlinear coupling waves, as noted in the former section.

       Notably, similar to the 'violent waves' recorded during this ship accident, the study of the Suwa-Maru incident (Tamura et al., 2009) also involved a sea state with co-existing swell and wind sea. The wave angle was $45°$, and the ratio of the two peak

frequencies was 0.92. In the Suwa-Maru incident, the narrow wave spectrum where the swell and wind sea co-occurred was the central argument for wave energy transformation because the oblique angle and close wave period conditions provided the optimal situation for wave coupling and energy interactions (Masson, 1993). Consequently, freak waves, which exhibited 20-fold higher nonlinear energy transfer compared with the original sea state, were generated and became the main threat to the safety of the ship.

Note that the wave model used in the present study is not tailored for special cases but rather is intended for global use. Therefore, the model produces differences when describing realistic sea states, especially because we observed that the swell was not particularly high during the accident. Nevertheless, a comparison of the sea state conditions in this case with those of the previously reported Suwa-Maru incident suggests that the oblique angles (especially from $40°$ and $60°$) and approaching wave periods of the swell and wind sea were the major factors that produced the 'violent waves' that consequently led to the

accident.

       In another case, a bulker carrier with a gross tonnage of 36,546 sailed from Davant, United States, to Hamburg, Germany, on 10 January 2010 at 14:45 UTC and encountered extremely poor weather, with westerly winds of more than 20 m/s and southwest waves of more than 9 m. The ship was damaged, and the crew was seriously injured at $46°14'$ N, $41°29'$ W. The sea state at this point was relatively high. From the perspective of wave height, the modeled $H_s$, $H_{ws}$ and $H_{sw}$ values between 12:00

UTC and 15:00 UTC increased from 8.08 m to 8.66 m, from 7.72 m to 8.16 m and from 2.34 m to 2.90 m, respectively. In addition, the wave periods increased from 10.5 s to 12.1 s for the swell and from 10.7 s to 11.1 s for the wind sea over three hours at this site. Evidently, the rising rate of the swell period was higher than that for the wind sea period, which produced a contour line of 1 s in the $\Delta T$ graphs. Concurrently, the waves in this case can be divided into two distinct areas according to the wave directions. The high $\Delta D$ area and low $\Delta D$ area were located in the northerly and southerly directions, respectively. The

boundary of the two areas at $50°\sim60°$ of $\Delta D$ was close to the accident area, thereby reflecting the fluctuating propagation angle and interactions between the two waves. These features, which are related to changes in the wave direction, were identical to those of the first case described above. In the present case, high waves (approximately 9 m) may have been a factor that threatened shipping activities. However, the decisive causes of the accident were likely related to the decreasing wave period and wave direction changes due to the co-occurring wind sea and swell conditions.

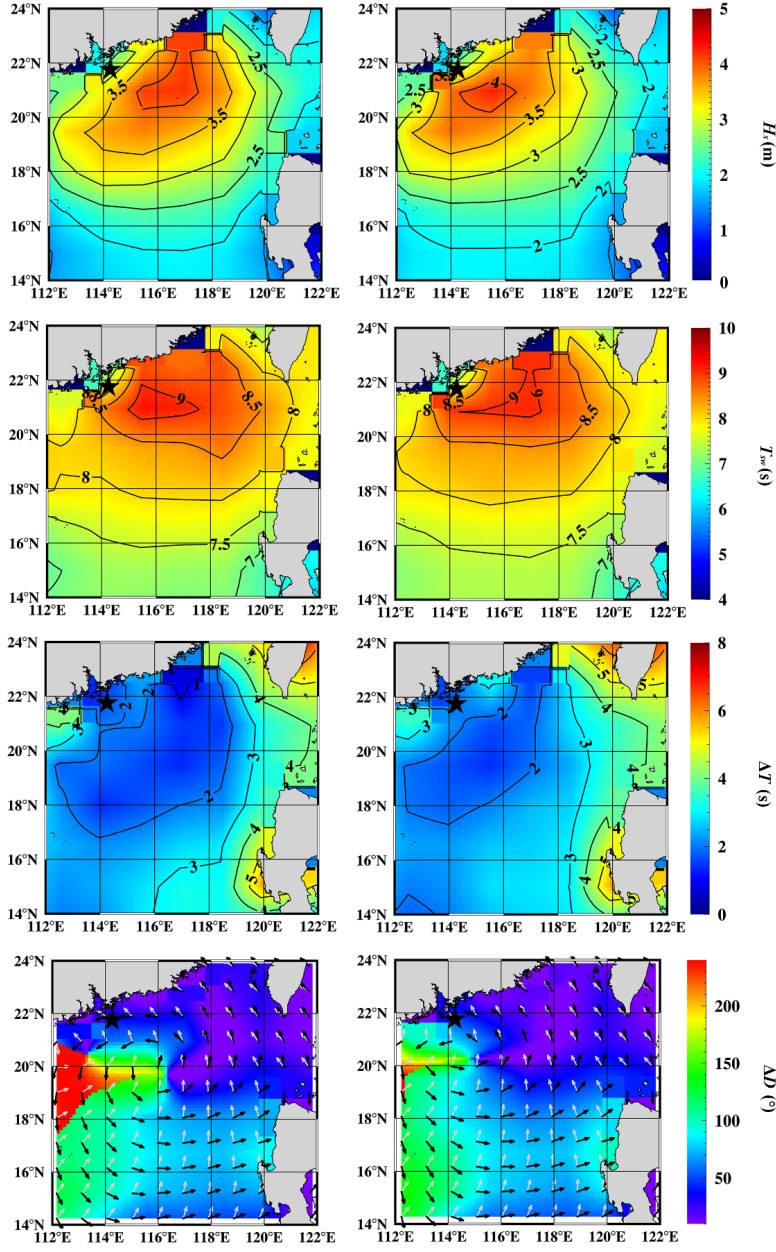

Figure 4. Sea states of Case 1 over 3 hours during the accident: $H_s$ (first row), $T_{sw}$ (second row), $\Delta T$ (third row) and $AD$ (fourth row). The first column shows the data for the sea states 3 hours before the accident, and the second column shows the data at the time of the accident. The accident point is represented by a black star, and the arrows in the fourth row represent the wave directions. The swell is indicated in black, and the wind sea is indicated in light grey. The figure was generated by the Interface Description Language (version 8.5) [software], Exelis Visual Information Solutions, Inc., a subsidiary of Harris Corporation (Exelis VIS) http://www.harrisgeospatial.com/ProductsandSolutions/GeospatialProducts/IDL.aspx.



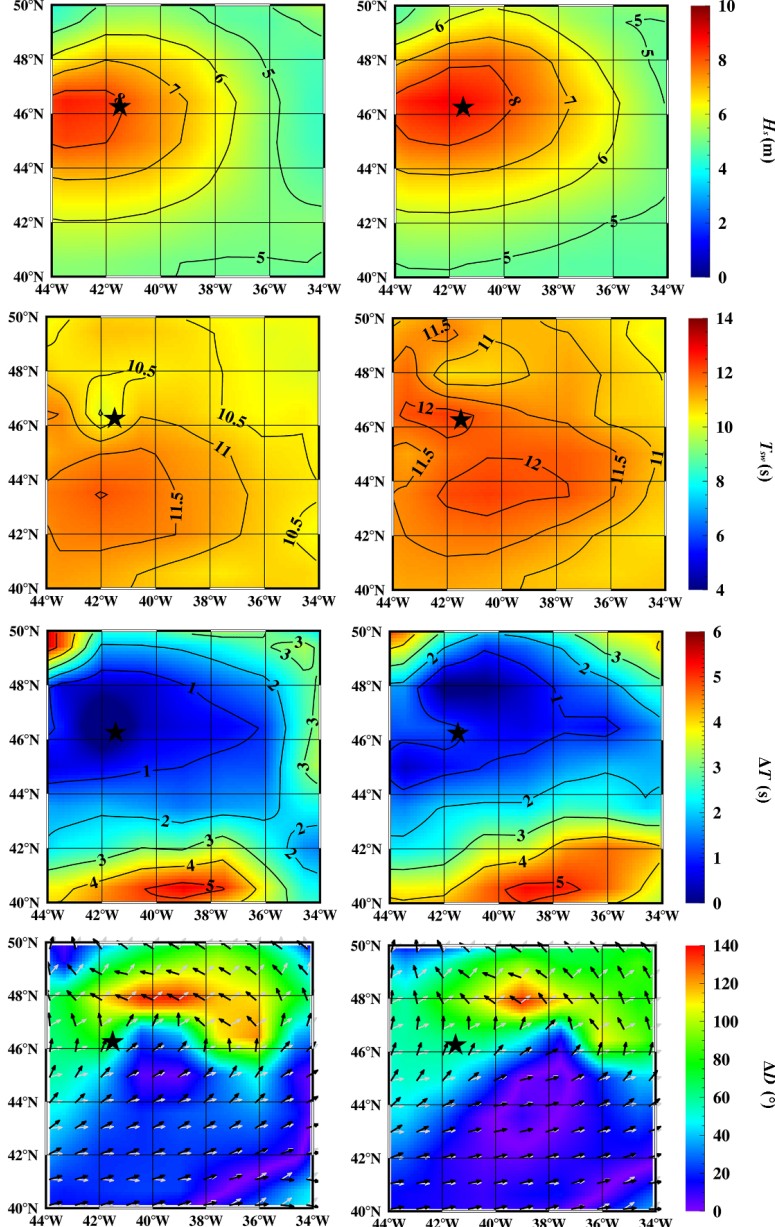

Figure 5. Sea states of Case 2 over 3 hours during the accident: $H_s$ (first row), $T_{sw}$ (second row), $\Delta T$ (third row) and $\Delta D$ (fourth row). The first column shows the data for the sea states 3 hours before the accident, and the second column shows the data at the time of the accident. The accident point is represented by a black star, and the arrows in the fourth row represent the wave directions. The swell is indicated in black, and the wind sea is indicated in light grey. The figure was generated by the Interface Description Language (version 8.5) [software], Exelis Visual Information Solutions, Inc., a subsidiary of Harris Corporation (Exelis VIS) http://www.harrisgeospatial.com/ProductsandSolutions/GeospatialProducts/IDL.aspx.





**6 Summary and Discussion**

The present study is motivated by a desire to thoroughly evaluate the sea state conditions during ship accidents. It aims to establish more accurate and effective maritime warning criteria and better understand the mechanisms underlying extreme waves. To this end, ten years of ship accidents that occurred in rough weather or sea conditions were chosen from the IMO ship
accident database and then analyzed.

Based on the selected 755 weather-related accident cases, an accident occurrence density map was generated. The ship accidents presented a dense distribution in the North Atlantic Ocean, the North Indian Ocean and the West Pacific Ocean because of the associated severe weather and sea state conditions, and the locations of these accidents coincided with major shipping routes. In terms of ship type in the casualty reports, the most frequent ships involved in these accidents included
general cargo ships, bulkers and fishing vessels. Of the reported initial events, stranding/grounding and hull damage were the most prominent.

Strong winds and high waves can cause heavy sea states, which are indeed the primary risk factors for maritime activities. However, the potential dangers of swells with relatively low wave heights are generally underestimated. Notably, our analysis of the 58 swell-related accidents indicated that 52% of the cases occurred in relatively low sea state conditions with $H_s$ values
smaller than 3 m, and the swells provided the dominant wave energy in these conditions. A further analysis of these accidents suggested that co-occurring wind sea and swell, especially when certain conditions occur, may lead to hazardous seas and pose a risk to shipping activities.

Our analysis of the wave period and wave direction demonstrated that two types of sea states can generate severe sea states. In the first state, $\Delta D$ is less than 30°, and the values of $\Delta T$ are all slightly great than 3 s. This case is the most likely angle
interval for two coupling waves to establish an extreme sea state (Onorato et al., 2010) because this scenario produces a large rate of wave amplification and generates unstable wave surfaces and perturbations. Although close wave period conditions are not present, the small difference angle between two wave systems has the strong possibility of forming violent seas. In the second state, as $\Delta D$ increases, $\Delta T$ reaches a minimum value of 1.8 s at 30° ~ 40° and is still less than 3 s when $\Delta D$ is large than 40°. Additionally, the nonlinear coupling between two waves reaches a high level as $\Delta D$ increases. Meanwhile, the
quasi-resonance occurs between two waves due to the close wave period. Therefore, with increasing wave energy transformation, waves can grow rapidly, become violent and threaten ships. Consequently, the resonant nonlinear interaction will result in a narrow wave spectrum and likely lead to an abnormal sea state under the influence of modulation instability. In addition, the rising value of wave steepness as $\Delta D$ increases indicates that the situation is worsening.

The dynamic wave interactions presented in the two specific cases analyses demonstrated that the oblique wave directions
(40° ~ 60°, listed in the cases) and the narrow wave periods between wind sea and swell led to the increase in the wave height, which could be an indicator of wave energy transformation and worsening sea state. This result is consistent with the explanation given in the previous paragraph.

Although the accuracy of the model data were validated using ERA-40 and operational archive results (Poli et al., 2013),





the present wave model results were retrieved on a global ocean scale rather than on an individual basis; therefore, deviations may exist compared to the actual sea states. Nevertheless, the results indicate noteworthy characteristics of dangerous sea state conditions.

Finally, ship safety could be improved if the major contributors to dangerous sea states are identified and monitored,

especially in high incidence areas for ships. In future work, the use of multisource data will undoubtedly provide a more complete description of these complex phenomena.

**Acknowledgments**

This study was partially supported by grants from the National Natural Science Foundation of China (No. 41406198), the "Pioneered Hundred Talents Program, Chinese Academy of Sciences" and the Hainan Key S&T Programme (No.

ZDKJ2016015). Additionally, we acknowledge that the global ship accident database and the ERA-20C Ocean Wave dataset were accessed from the IMO Global Integrated Shipping Information System (https://gisis.imo.org/Public/Default.aspx/) and the ECMWF Public Datasets web portal (http://apps.ecmwf.int/datasets/data/era20c-wave-daily/type=an/), respectively.

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
