# Peer review of "Global ship accidents and ocean swell-related sea states"

_Natural Hazards and Earth System Sciences, 2017_

## Referee Comment (RC1) · T. Bruns (Referee) · 8 May 2017

1. General comments It is an interesting paper investigating the correlation between ship accidents and certain sea states, co-occurrence of wind sea and swell, in particular. It is well known that ships are endangered in high seas, depending on their size. Therefore, the IMO recommends ship owners to take advantage of ship routing services. However, accidents also occur unexpectedly at moderate sea states and thus it makes sense to dig deeper in the available data base. From this point of view, I find this paper is worth being published.

However, I have some major concerns and proposals of improvement: • The wave model data set ERA-20c used is not suitable for this study, therefore acquire a high resolution data set • Discuss the typical cases in terms of time series in addition to

the 2-dim-representation • Include interpretation of wave spectral partitioning • Include some discussion on ship behavior in rough seas

Under these conditions , the paper will be acceptable , otherwise, I would have to reject the paper.

My comments in detail: 2. Numerical Wave Model Data :

Page 3, line 24-25: The Ocean Wave Daily data in the ERA-20C dataset are available from 1900–2010 every 3 hours at a grid size of $0.125°$ERROR $\rightarrow$ The spatial resolution is actually $1.5°$ !! The coarseness is obvious in figures 4 and 5. Near coastal wave heights tend to be underestimated. Referring to section 4, most of the ship accidents occurred near coasts!

3. Overview of Ship Accidents

The section describes the large variety of ship types and accidents. It is clear that each case would deserve a distinct study taking into account the ship's properties. Since this will hardly be possible, some discussion on ship's behavior in heavy seas would be helpful at this point. The authors already mention parameteric rolling, but a lot other dangerous incidents may occur, like extreme slamming, bending and torsional stresses, green water on deck and emerging propellers (both reducing ship's stability).

4. Analysis of the Sea State during Ship Accidents With the statistical evaluation in this section the authors attempt to find evidence of a relation between ship accidents and crossing seas (wind sea and swell), characterized by small $\Delta T < 2s$, directional spread of 30-40°. Indeed, a tendency in this direction can be recognized, but there is a lot of noise in the data, which remains unexplained. Some portion of this noise is probably due to the coarse data grid. The rest should be explainable by the large variety of ships and different kinds of causes for accidents (see 3.). Human errors may also cause accidents, even in moderate sea states. A discussion of this issue should be included here. Wave steepness values between 0.03 and 0.04 are surely

not hazardous for ships at all, but steepness of individual waves will probably increase the average steepness. The positive correlation between steepness and the spread between sea and swell propagation is interesting: Is there some theoretical explanation for this phenomenon?

5. Sea States of Typical Cases It should be mentioned that the distinction between wind sea and swell is an artificial construction. Mariners usually think of swell as a wave system originating from a distant storm, travelling into the local wind field. Crossing seas of this type are not very frequent but very hazardous. The sea states discussed in this section and presumably most of the other 753 cases do not involve a "classical" swell. Crossing seas with angles of 30-40° between directions of wind sea and swell are typically generated by rapidly moving low pressure systems, particularly in the vicinity of cold fronts with sudden changes in wind direction. Consider a storm with high wind sea : If the wind changes direction, a new wind sea is generated and the "old" wind sea is transformed to "swell". This partitioning of wave systems is done by the wave model post processing.

Figures 4 and 5 are difficult to interpret, I could hardly follow the arguments in the text. The coarse 1.5° model grid creates strange rectangular wiggles of contours. Furthermore, I miss date and time in the legend. In order to thoroughly investigate the cause of accidents I suggest a local description in terms of time series like the ones I include here, based on the operational ECMWF Global WAM. Note that in case 1 waves are twice as high as in ERA-20c! Obviously, this is not a "low sea state" case.

6. Minor revisions Page 2, line 5 : A high wave height is no doubt undoubtedly a threat 5 for ships, yet some ships wreck at relatively low wave heights and but high wave steepness sea states (Toffoli et al., 2005). Page 2, line 28-29 : The detailed information discussed above are is presented in section 2. Page 3, line 22: "ERA-Interim" has not been described so far. Page 5, line 1: The number of ship accidents that fell within inside each region was summed . . .

Page 6, line 2: As discussed in the Introduction section, the co-occurrence of wind sea and swell conditions is considered a potential causal.

Page 8, line 4: a small $\Delta D$ area in the northerly direction and a large $\Delta D$ area in the southerly direction. $\rightarrow$ seen from the ship's position? This is hardly recognizable on the map of fig 4!

Please also note the supplement to this comment:
http://www.nat-hazards-earth-syst-sci-discuss.net/nhess-2017-142/nhess-2017-142-RC1-supplement.pdf

[Figure]

[Figure]

**Fig. 1.** Time series of wind and waves in case 1

[Figure]

**Fig. 2.** Time series of wind and waves in case 2

---

## Author Comment (AC1) · 9 Jun 2017

We would like to thank Dr. Bruns for his valuable comments on the manuscript. While we agree with most of the comments, we think that the ERA-20C data should be appropriate for us to conduct this study.

We are not a state member of ECMWF and therefore it is difficult for us to acquire the operational WAM model data run by the ECMWF, as the example shown by the referee. Considering that we investigated impacts of both windsea and swell on ship accidents occurring in a period of ten years, and ERA-20C publicly provides both the "partitioned" windsea and swell parameters for a long period, we decided to use such dataset. When one downloads the ERA-20C dataset, data with various spatial resolution (from the lowest 3° by 3° to the highest 0.125° by 0.125°) are available. We used the ERA-20C data with the highest resolution of 0.125° by 0.125° for this study. Fig.S1 in the supplementary shows the screenshot of the data downloading interface (at http://apps.ecmwf.int/datasets/data/era20c-wave-daily/type=an/)

Although the ERA-20C data that we used in this study has a good spatial resolution, it may have bias in some special cases, as in the case shown in our manuscript. In the following, we presented a further detailed analysis for the first case, i.e. the Chicago Express accident case occurred on Sep.23, 2008.

In the preparation of the manuscript, we just found it's an interesting case as this is a typical cross sea situation, whereas we didn't recall this one was actually a very complicated ship accident case, not only high winds and waves but also possible parametric rolling occurred due to the severe weather induced by the typhoon "Hagupit". As recommended by the referee, time series of sea state parameters are depicted for a clearly discussion (see in Fig.S2 in the supplementary). Indeed, distinct bias exists on the other parameters from ERA-20C data, particularly SWH of the windsea (which of course leads to bias of SWH of total sea) comparing to the operational WAM model data (seen from the figure provided by the referee).

Meanwhile, we also further validated the ERA-20C results of this case by comparing with radar altimeter measurements. The ENVISAT RA-2 measurements were available between 13:40 -14:00 in the vicinity of the accidents on August 23. The RA-2 shows that the SWH was higher than 7 meters in the open sea of Hong Kong, whereas the ERA-20C model data only yields SWH of approximated 4-5 meters.

By comparing the ERA-20C model sea surface wind data with the Scatterometer measurements, we found that the ERA-20C underestimated the sea surface wind, as shown in Fig. S4, and it didn't depict well the center of the typhoon. At 12:00 UTC on August 23, the highest sea surface wind speed of ERA-20C in the Hong Kong open sea was only approximately 20 m/s (Fig.S4(a)), whereas the ASCAT measurement shows that

the sea surface wind speed was higher than 25 m/s in the typhoon center (Fig.S4(b)). Although the temporal interval between the ERA-20C and the ASCAT measurement is about one and half hour, the comparison suggests that the ERA-20C didn't depict well the weather situation during this typhoon process. As the sea surface wind speed is significantly underestimated in ERA-20C, it is not strange that wave height of windsea is consequently underestimated.

Above the general situation of this typhoon case is presented. A detailed analysis of this Chicago express ship accident actually is published by the Bundesstelle fuer Seeunfalluntersuchung (Federal Bureau of Maritime Casualty Investigation, Germany). The document is titled "Fatal accident on onboard the CMV CHICAGO EXPRESS during Typhoon "HAGUPIT" on 24 September 2008 off the coast of Hong Kong". In this document, a detailed analysis of sea state compiled by the experts (perhaps by Dr. Bruns and his team) from the DWD (Deutsche Wetterdienst) is presented. One can find this document in: http://www.bsu-bund.de/EN/Publications/Unfallberichte/_functions/unfallberichte_table_2009.html.

Overall, given that the special situation of the typhoon Hagupit leads to this ship accident, this case is not an appropriate one for discussion in this paper. A better one has been selected for the further study, which will be presented in the revision of this manuscript. Meanwhile, we are working on the validation of ERA-20C data by comparing with radar altimeter measurements for the selected cases.

This short response is compiled by Dr. Xiao-Ming Li and Ms. Zhiwei Zhang for further open discussion. A point-by-point response to the referees' comments will be provided in the revision. We would like to thank Dr. Bruns again for pointing out that the inappropriate case analysis.

Please also note the supplement to this comment:
http://www.nat-hazards-earth-syst-sci-discuss.net/nhess-2017-142/nhess-2017-142-AC1-supplement.pdf

---

## Referee Comment (RC2) · Anonymous Referee #2 · 13 Jun 2017

This manuscript focuses on the relation between swell and ship accident. The motivation is very interesting and the result is reasonable. Under this condition, I recommend it should be published. However, several mistakes has to be fixed before its publication.

Abstract: -sea state conditions play 10 a significant role in shipping safety. should be sea state. remove the condition

-The sea state parameters, including the significant wave height, the mean wave period and the mean wave direction, obtained from numerical wave model data were analyzed for selected ship accidents. This sentence is fuzzy.please rewrite it

Introduction -wave period (T) cross-zero wave period or other types of wave period?

Date and method -the ERA-20C products describe the spatio-temporal evolution of the

atmosphere (on 91 vertical levels, between the surface and 0.01 hPa), the land-surface (in 4 soil layers), and ocean waves (for 25 frequencies and 12 directions). I understand you want to describe the high-quality of ECMWF-20C data, however, waves are used in this study. Thus the atmosphere and soil are useless here.

-I wonder why not analysis the relation between ship accident and winds? especially in poor weather, wave should be related with wind.

-4.1 Wave Height Here, the variable is not coincident with description above. Following the manuscript, SWH is right!

-Figure 4 the figure at forth row should be replotted due to the colors does not overlap with x axis and y axis

-Figure 5 the arrow is out of area.

---

## Author Comment (AC2) · 10 Jul 2017

We would like to thank the reviewer for the valuable comments on the manuscript. Following are responses to the comments on a point-by-point base.

C1: Abstract: -sea state conditions play a significant role in shipping safety. should be sea state. remove the condition.

R1: The suggestion will be followed in revision of the manuscript.

C2: The sea state parameters, including the significant wave height, the mean wave period and the mean wave direction, obtained from numerical wave model data were analyzed for selected ship accidents. This sentence is fuzzy. please rewrite it.

[Figure]

R2: The sentence is revised: Sea state parameters of numerical wave model, in terms of significant wave height, mean wave period (zero-crossing period) and mean wave direction were analyzed for the selected ship accident cases.

C3: Introduction -wave period (T) cross-zero wave period or other types of wave period?

R3: Yes. It is cross-zero wave period. We will clarify this in the revised manuscript.

C4: Date and method -the ERA-20C products describe the spatio-temporal evolution of the atmosphere (on 91 vertical levels, between the surface and 0.01 hPa), the land-surface (in 4 soil layers), and ocean waves (for 25 frequencies and 12 directions). I understand you want to describe the high-quality of ECMWF-20C data, however, waves are used in this study. Thus the atmosphere and soil are useless here.

R4: Yes. The sentence is a little bit lengthy and we will revise it accordingly.

C5: I wonder why not analysis the relation between ship accident and winds? especially in poor weather, wave should be related with wind.

R5: It is a good question. Sea wind does have significant impact on shipping safety and in many cases the high waves induced by wind can cause serious ship casualties. However, in this study, we would like to know impacts of sea state when both windsea and swell present on shipping safety. Swell are long waves propagating far away from generation sources and therefore are not effected by sea wind anymore. Therefore, in this study, we didn't investigate relation between sea wind and ship accidents. We will clarify this point in the revised manuscript.

C6: 4.1 Wave Height Here, the variable is not coincident with description above. Following the manuscript, SWH is right!

R6: We will follow this suggestion and revise the subtitle.

C7: Figure 4 the figure at forth row should be replotted due to the colors does not

overlap with x axis and y axis

R7: Thank the reviewer for pointing the slight offset of the two plots in the fourth column. We will revise them.

C8: Figure 5 the arrow is out of area

R8: Starting points of the arrows are the grids where the model data available. The boundary (axis limits) of the plots are the same as the grids, therefore, it is unavoidable that some arrows are beyond the plots.

---

## Author Response (AR1)

**Global ship accidents and ocean swell-related sea states**

Zhiwei Zhang[1, 2], Xiao-Ming Li[2, 3, 4]

[1] East Sea Information Center, State Oceanic Administration, Shanghai, China
[1,2] College of Geography and Environment, Shandong Normal University, Jinan, China
[2,3] Key Laboratory of Digital Earth Science, Institute of Remote Sensing and Digital Earth, Chinese Academy of Sciences, Beijing, China
[3,4] Hainan Key Laboratory of Earth Observation, Sanya, China

*Correspondence to:* X.-M. Li (E-mail: lixm@radi.ac.cn)

**Abstract.** With the increased frequency of shipping activities, navigation safety has become a major concern, especially when economic losses, human casualties and environmental issues are considered. As a contributing factor, the sea state  plays a significant role in shipping safety. However, the types of dangerous sea states that trigger serious shipping accidents are not well understood. To address this issue, we analyzed the sea state characteristics during ship accidents that occurred in poor weather or heavy seas based on a ten-year ship accident dataset. Sea state parameters of a numerical wave model, i.e.  significant wave height,  mean wave period and  mean wave direction,  were analyzed for the selected ship accident case. The results indicate that complex sea states with the co-occurrence of wind sea and swell conditions represent threats to sailing vessels, especially when these conditions include  similar wave periods and oblique wave directions.

**1 Introduction**

The shipping industry delivers 90% of all world trade (IMO, 2011). It is currently a thriving business that has experienced increases in both the number and size of ships. However, due to the frequency of shipping activities, ship accidents have become a growing concern, as have the  associated destructive consequences including: casualties, economic losses and various types of environmental pollution.

Investigations into the causes of shipping accidents show that over 30% of the accidents are caused by poor weather, and an additional 25% remain completely unexplained (Faulkner, 2004). Due to these dangerous uncertainties, accidents that involve poor weather and severe sea states should be further studied for shipping safety.

However, under changing weather conditions, the sea surface is too complex to predict, especially on  short timescale (Kharif et al., 2009). The sea surface is composed of random waves of various heights, lengths and periods. Meanwhile, different kinds of waves emerge frequently among them, wind sea and swells are the two main types of ocean waves classified by wave generation mechanisms. Wind sea waves are directly generated by local winds, and when wind-generated waves propagate without receiving further energy from wind,  they transition in swell.

Meteorologists and oceanographers generally work with statistical parameters, such as the significant wave height ($H_s$),

wave period (*T*, zero-crossing period) and wave direction (*D*), to represent a given sea state. Additionally, the wave spectrum, i.e. the distribution of wave energy among different wave frequencies ($f, f = 1/T$) is analyzed in some studies to better understand wave dynamics. Note that a typical ocean wave spectrum with two peaks (e.g., one from  distal swell and the other generated by the local wind) are much more complicated and variable.

5       In terms of the sea state parameters, $H_s$ is usually a practical indicator of the sea state during marine activities. Indeed, some studies, $_s$ such as an analysis of ship accidents that occurred in the North Atlantic region (Guedes et al., 2001), have shown that accident areas coincide with the zones with the highest $H_s$. A high wave height is  undoubtedly a threat  to ships, yet some ships wreck  in sea states characterized by relatively low wave heights and high wave steepness  (Toffoli et al., 2005).

10      A sea state with a narrow wave spectrum was observed during several major ship accidents, including the 'Voyager' accident (Bertotti and Cavaleri, 2008), the Suwa-Maru incident (Tamura et al., 2009), the Louis Majesty accident (Cavaleri et al., 2012) and the Onomichi-Maru incident (In et al., 2009; Waseda et al., 2014). Studies have assumed that the narrowed wave spectrum is primarily generated by the nonlinear coupling of swell and wind sea (or swell and swell) (Bertotti and Cavaleri, 2008; Tamura et al., 2009; Cavaleri et al., 2012; Waseda et al., 2012). During such wave couplings, the wave energy from one wave system (wind sea or swell) is enhanced and  transferred to the other wave system (usually a swell) (Tamura et al., 2009; Waseda et al., 2014). As a result, the wave energy transformation produces a steep swell, with a  high wave energy and extreme wave height (Bertotti and Cavaleri, 2008).

     The oblique angle between two waves is another important condition involved in the interaction of wave systems.  The traveling angles  associated with ship accidents  have varied from 10° (Onorato et al., 2010) to 60° (Tamura et al., 2009). The features noted above emerge individually or simultaneously  during ship accidents or  rare extreme sea states when swells and wind seas co-occur. Indeed, the co-occurrence of wind seas and swells can lead to dangerous seas, as demonstrated by the parametric rolling  experienced by the German research vessel Polarstern (Bruns et al., 2011),  despite the absence of extreme wave heights .

     In previous studies of ship accidents, researchers focused on only one severe accident when discussing the sea state dynamics in detail or based their studies on ship accident data to perform statistical analyses of classical sea state parameters (e.g., $H_s$ and *T*). To thoroughly investigate sea state parameters, we collected information on a large number of ship accidents and created a database for analysis. Additionally, we discussed the parameters in both wind sea and swell conditions. Statistical analyses were performed on data obtained from the International Maritime Organization (IMO). The data include ten years of ship accidents (2001 – 2010) and 755 cases caused by bad weather or heavy seas. Because swells with large wave energies can represent a threat to maritime activities, 58 cases in which swells were reported as an important factor in the ship accident were selected. The detailed information discussed above  is presented in section 2. Following an overview of the ship accidents (section 3), an analysis of the swell-related sea state conditions for these ship accident cases is presented in section 4. In section 5, two cases are illustrated to demonstrate the dynamic processes that ensue when wind sea and swell conditions occur during ship accidents. Finally, a summary and discussion are provided (section 6).

**2 Data and Methods**

**2.1 Ship Accident Database**

A ten-year (2001–2010) ship accident dataset was gathered from the Marine Casualties and Incidents Reports issued by the IMO. The dataset includes 3648 ship accidents, and each accident in the report includes the occurrence information, such as the accident time and coordinates, initial event, summary, casualty type, ship type, among other factors. Since the primary information used in this study includes the accident time and coordinates,  events that failed to record these details were excluded, and 1561 cases with exact geographical locations remained in the dataset.

According to the description of initial events, which provides clues regarding the accident causes recorded in the reports, those 1561 valid cases cover different kinds of cases triggered by natural factors and human factors. Because we focus on the events that occurred in natural weather-related conditions, cases with  descriptions such as fire or explosion, improper operations, and lost persons were eliminated from the 1561 cases, while cases with keywords such as strong wind/gale/cyclone or heavy seas/rough waves were kept. Although the proximal human factors resulting in ship accidents recorded in the IMO reports may have been indirectly related to dangerous seas or heavy weather, e.g., improper operations by crews, it would be exceedingly difficult to analyze the original factors case by case. Thus, distinguishing among trigger factors based on initial event keywords represents an optimal way of filtering the dataset. After this filtering, 755 weather-related accidents were obtained for the further analysis. An overview of these 755 cases is presented in section 3.

Furthermore, this study focuses on the cases that occurred in swell-related sea states. After examining all the summaries of the 755 cases, we  retained 58 cases with clear descriptions of the swell  motion during the ship accidents for the analysis of the swell-related sea states.  A detailed analysis is presented in section 4.

**2.2 Numerical Wave Model Data**

The ERA-20C numerical wave model data were obtained from the European Center for Medium-Range Weather Forecasts (ECMWF). The ECMWF uses atmosphere, land, surface, and ocean wave models and data to reanalyze the weather conditions during the last century. The ERA-20C products describe the spatio-temporal evolution of  ocean waves for 25 frequencies and 12 directions. The accuracy is improved by validation with ERA-40 data and operational archive results. Compared to the ERA-Interim dataset (12 h), ERA-20C has longer reanalysis coverage (24 h) for single-point data (Poli et al., 2013). The Ocean Wave Daily data in the ERA-20C dataset are available from 1900–2010 every 3 hours at a grid size of 0.125°. The data provide 33 reanalyzed ocean wave parameters, and separate entries are included for swell and wind sea .

**3 Overview of Ship Accidents**

In the ship accident dataset, 755 weather-related cases were distinguished and  discussed in section 2. Hereafter, we

provide an overview on these 755 cases in terms of the initial events, ship types and spatial distribution.  The initial event in the IMO reports describes the triggering behaviors of each accident. Based on these records, five types of initial events were selected for

5    classification, which are stranding/grounding, hull damage, others, capsizing/listing and foundering/sinking, sorted from the largest proportion to the smallest (Figure 1(a)). The initial event labeled others in the classification includes report keywords such as 'machinery damage due to heavy weather', 'cargoes shifting due to rough seas' and 'fatalities in heavy weather conditions', which are all related to bad weather. Note that the classification shown in Figure 1 (a) is not based on a detailed trigger factor but a general result. For instance, when the ship accidents are classified as stranding/grounding or

10   foundering/sinking, the vessels may have suffered from various types of dangerous seas or bad weather, including parametric rolling, extreme slamming, bending and torsional stresses, and/or green water on deck, all of which all can reduce a ship's stability and consequently cause stranding/grounding or sinking.

     Different types of ships respond differently when they encounter potentially dangerous sea conditions because of their  different structures and functions. Among the 755 cases, general cargo vessel types experienced the

15   highest proportion of accidents (32.3%) in rough weather and severe sea states,  followed by bulkers and fishing vessels. Collectively, these data highlight the types of ships that may require more attention during shipping activities (Figure 1(b)).

[Figure]

[Figure]

(a) Classification based on initial events       (b) Classification based on ship types

[revised manuscript text omitted]

The first ship accident case occurred at approximately 20:30 UTC on February 24th, 2009. The Korean tug CHONG JIN

capsized at 34°8′ N, 124°131′ E. To thoroughly investigate the possible cause of this accident, the sea state is analyzed in detail. Figure 4 shows the time series of the sea state and sea surface wind at the accident location over 24 hours. At the top of the graph, wind vanes and numbers represent the sea surface wind direction and wind speed. The lines in the middle of graph represent significant wave heights of the swell (blue), wind sea (green) and total sea (grey), respectively. The mean wave period of the swell and wind sea are annotated in the same colors as the wave height. The wave directions of the swell and wind sea are also presented at the bottom of the graph.

[Figure]

**Figure 4. Time series of sea surface wind and sea state at the ship accident location over 24 hours for the case occurred at 20:30 UTC on February 24th, 2009. At the top of the figure, wind vanes and numbers indicate wind direction and wind speed at the accident location. Three polylines that in grey, green and blue colors represent the significant wave height of total sea, wind sea and swell, respectively. The numbers in green and blue colors are the mean wave period of wind sea and swell. At the bottom of the figure, the arrows that in green and blue color indicate the mean wave direction of wind sea and swell, respectively.**

At 12:00 UTC on February 24th, approximately 8 hours before the ship accident, the sea was low with an $H_s$ of 0.7 m. The dominant wave was swell moving to the northwest, and the wind sea was fairly slight. At 15:00 UTC, the wind sea began to develop rapidly due to sudden changes in the wind field (the wind direction changed from east to northwest, and the wind speed increased from 3.2 m/s to 8.4 m/s). Along with the continual growth of the wind sea, the difference between $T_{sw}$ and $T_{ws}$ decreased. The wave directions of the wind sea and swell at this moment were almost opposite. Soon thereafter, at 18:00 UTC, the wind speed rose continuously and reached 12.7 m/s, while the wind direction tended to the north. The $H_{ws}$ reached 1.4 m, while the $H_{sw}$ was still less than 1 m. When the accident occurred (close to 21:00 UTC), the swell direction was distinctly different from that at 18:00 UTC, as it had shifted from southeasterly to northwesterly. As the sea became rough, both $H_s$ and $H_{ws}$ increased rapidly by 2 m. Similar growth occurred in the wave periods, specifically, from 5.7 s to 6.2 s for the swell and from 4.3 s to 5.4 s for the wind sea. As the wave period of the wind sea became close to the swell, the difference

between them ($\Delta T$) decreased to 1 s.

Figure 4 shows variations in the sea state at the accident location, while sea state in the vicinity of the accident is presented in Figure 5. The diagrams in the first column show the wave model results at 18:00 UTC, 3 hours before the accident, whereas the second column are the results at 21:00 UTC, close to the accident occurrence time. From top to bottom, the sea state parameters are $H_S$, $\Delta T$ and $\Delta D$. Within three hours of the ship accident occurrence, the $H_S$ increased slightly by approximately 0.5 m across a large area proximal to the accident location. Due to the growth of the wind sea forced by local wind (refer to Figure 6), the wave period difference $\Delta T$ decreased by approximately 2 s in the area. The most distinct variation in sea state in the area is $\Delta D$. At 18:00 UTC, the swell direction in the area was southeasterly, propagating from other areas to the ship accident location. However, after three hours, the swell direction changed to northwesterly, opposite the direction three hours before. Based on the wave direction of the wind sea in the area, we find that the northwesterly swell system (at 21:00 UTC) likely transformed from a fully developed local wind sea after the wind direction turned to the north at 15:00 UTC. Consequently, $\Delta D$ narrowed greatly from 187° to 59° at 21:00 UTC, further decreasing to 30° at 3:00 UTC on February 25th.

The change in the wind is considered a key factor in this accident. The turning point appeared at 15:00 UTC on the 24th, when the sea wind changed significantly in terms of both magnitude and direction. Afterwards, the continuous force of the sea wind induced the growth of the wind sea. The "old" wind sea was transformed into a young swell, which led to a marked decrease in $\Delta D$. The closer wave period and narrower direction angle between the wind sea and the swell produced a resonance effect. Some experimental studies suggest that a swell with the same direction as the wind will play a role in suppressing the growth of a wind sea (Philips and Banner, 1974; DoneJan, 1987). As the swell direction tends to be the same as the wind direction, the development of a wind sea is suppressed. Meanwhile, the lower $\Delta D$ and $\Delta T$ provided conditions for wave energy transformations from the wind sea to the swell (Masson, 1993). Closer wave periods and narrower wave spectrum provide ideal conditions for the transformation of wave energy, and the resulting energy-enhanced swell represents a great threat to shipping safety.

Based on the analysis presented above, we found that the crossing sea state of swell and wind sea may have triggered the accident. Moreover, the swell that had a significant impact on the ship accident was transferred from the local wind sea instead of the "old" one that propagated from a distant storm. In the numerical wave modeling, discrimination of "swell" and "wind sea" occurs in the post-processing step through wave spectral partitioning. This spectral partitioning arbitrarily divides the two-dimensional wave spectrum into wind sea and swell components (Gerling, 1992; Hanson and Phillips, 2000) based on some criteria, and the integrated wave parameters of the corresponding wind sea and swell are subsequently derived. These swell and wind sea values are useful for depicting the trend of a sea state and can significantly contribute to many applications, such as forecast and analysis of surface wave conditions in shipping lanes and coastal areas, as in the statistical analysis and the case study presented above. However, spectral partitioning may have trouble distinguishing among the essential attributes of a wave field when both wind sea and swell or multiple swell systems are present (Hanson and Phillips, 2000).

[Figure]

**Figure 5. The sea state in the vicinity of the case that occurred on February 24th, 2009. The left and right columns are the model results at 18:00 and 21:00 UTC, respectively. The first, second and third rows in the figure are $H_s$, $\varDelta T$ and $\varDelta D$, respectively. The accident location is marked with a black star. Arrows in the plots of the third row represent the wave directions of the swell (black) and wind sea (light grey).**

[Figure]

**Figure 6. The sea surface wind fields at 18:00 UTC (left) and 21:00 UTC (right) of the first case that occurred at 20:30 UTC on February 24th, 2009. The accident location is marked with a black star. The arrows represent the sea surface wind directions.**

5    In  the second case, a bulker carrier with a gross tonnage of 36,546 sailed from Davant, United States, to Hamburg, Germany, on  January 10, 2010, and at 14:45 UTC  encountered extremely poor weather, with westerly winds of more than 20 m/s and southwest waves of more than 9 m. As a result, the ship was seriously damaged at 46°14′ N, 41°29′ W. The time series of the sea surface wind and sea state over 24 hours for this case are presented in Figure 7. The lines, symbols and numbers in the figure have the same meanings as those presented in Figure 4.

[Figure]

**Figure 7. Time series of sea surface wind and sea state at the ship accident location over 24 hours for the case that occurred at 14:45 UTC on Jan. 10th, 2010. The lines, symbols and numbers in the figure have the same meanings as those presented in Figure 4.**

[Figure]

**Figure 8. The sea state in the vicinity of the case that occurred on January 10th, 2010. The left and right columns are the model results at 18:00 and 21:00 UTC, respectively. The first, second and third rows in the figure are $H_s$, $\Delta T$ and $\Delta D$, respectively. The accident location is marked with a black star. Arrows in the plots of the third row represent the wave directions of the swell (black) and the wind sea (light grey).**

~~The sea state at this point was relatively high. From the perspective of wave height, the modeled $H_s$, $H_{ws}$ and $H_{sw}$ values between 12:00 UTC and 15:00 UTC increased from 8.08 m to 8.66 m, from 7.72 m to 8.16 m and from 2.34 m to 2.90 m, respectively. In addition, the wave periods increased from 10.5 s to 12.1 s for the swell and from 10.7 s to 11.1 s for the wind sea over three hours at this site. Evidently, the rising rate of the swell period was higher than that for the wind sea period, which produced a contour line of 1 s in the $\Delta T$ graphs. Concurrently, the waves in this case can be divided into two distinct areas according to the wave directions. The high $\Delta D$ area and low $\Delta D$ area were located in the northerly and southerly directions,~~

respectively. ~~The boundary of the two areas at 50°–60° of *ΔD* was close to the accident area, thereby reflecting the fluctuating propagation angle and interactions between the two waves. These features, which are related to changes in the wave direction, were identical to those of the first case described above. In the present case, high waves (approximately 9 m) may have been a factor that threatened shipping activities. However, the decisive causes of the accident were likely related to the decreasing wave period and wave direction changes due to the co-occurring wind sea and swell conditions.~~

Based on the sea surface wind field on January 10$^{th}$ and 11$^{th}$ over a large area in the vicinity of the ship accident (not shown here), the area was experiencing an extra-tropical cyclone. The $H_s$ (over 8 m) and sea surface wind speed (higher than 20 m/s) presented in Figures 7 and 8 also reveal the bad weather situation when the ship accident occurred. From 3:00 UTC until the approximate time of the ship accident, the wind sea grew under the force of the continuously increasing sea surface wind speed, as evidenced by the increases in $H_{ws}$ and $T_{ws}$. The time at which the ship accident occurred, i.e. approximately 15:00 UTC, likely corresponds to a turning point in the wind sea growth. Before then, the $H_{ws}$ continuously increased from 3.7 m at 3:00 UTC to 8.2 m at 15:00 UTC. Simultaneously, the $T_{ws}$ increased from 8 s to 11 s. After the turning point, both the wave height and wave period of the wind sea started to decrease gradually. More interesting is the variation in the swell in the area. At the ship accident location, both $H_{sw}$ and $T_{sw}$ gradually increased from 3:00 UTC until 21:00 UTC on January 10$^{th}$. However, the time series of the mean wave direction of swell shown in Figure 7 suggests that the swell situation was very complicated. At 3:00 and 6:00 UTC on January 10$^{th}$, the easterly swell was the dominant swell system. The mean wave direction of the swell subsequently gradually turned to southerly and southwesterly, leading to a decrease in the *ΔD* from 107° (3:00 UTC) to 59° (15:00 UTC).

The sea state maps shown in Figure 8 can better resolve the variations over the course of a few hours. The $H_s$ graph clearly shows that the higher wave area increased in size within 3 hours. The *ΔT* map suggests that the swell and wind sea had a close mean wave period of less than 1 s in a large area surrounding the ship accident location. In fact, the *ΔT* derived from the time series presented in Figure 7 suggests that it retained a quite small value of 0 ~ 1 s for approximately 15 hours. The *ΔD* graphs show that at least two swell systems existed in the area, with one being southerly/southwesterly and the other being southeasterly. This situation led to a high *ΔD* area and a low *ΔD* area. The boundary of the two *ΔD* areas at 50°~60° was close to the accident area. Based on both Figures 7 and 8, the *ΔD* was becoming smaller during the event, not only in the location of the ship accident but also across a large area. The time series of the swell wave direction may misleadingly suggest that the swell direction changed suddenly within a few hours. However, this was probably not the true situation. As stated in the analysis of the first case, discrimination of the swell and wind sea in the wave model post-processing step is an arbitrary process. The wave model product used in this study only provides one swell component, which cannot represent the complete swell state in a complicated situation. In this case, as mentioned above, the area was experiencing an extra-tropical cyclone that not only featured a rotational wind field but also moved in a certain direction, thereby generating swells propagating in various directions. Thus, multiple swell components might have co-existed in certain observation locations. However, spectral partitioning cannot resolve the complete swell components as only one (dominant) swell system was retained in the wave

product, such as that used in this study. This situation can lead to misunderstanding data suggesting that the swell direction changed suddenly.

Nevertheless, in the present case, large waves (higher than 8 m) may have been a factor that threatened shipping activities. The additional causes of the accident were likely related to the decreasing wave period and wave direction changes that led to co-occurring wind sea and swell.

**6 Summary and Discussion**

[revised manuscript text omitted]

According to the report records, many ship accidents have occurred in offshore areas yet few have occurred in open sea areas. Although the accuracy of the model data is fairly high, the coastline resolution used in the dataset is relatively coarse. As a result, a bias may exist in the offshore areas. Therefore, the diagrams shown in the statistical analysis appear to be somewhat noisy. On the other hand, even under the same sea state, different ship types should respond differently. This phenomenon may also lead to a variety of statistical results in a variety of situations. Nevertheless, the statistical results still reveal noteworthy characteristics of dangerous sea state conditions.

The sea states of the two case studies meet the general conditions of a possible occurrence of dangerous waves based on the statistical analysis, whereas they also presented different situations. In the first case, the overall sea state was relatively low, at 2.0 -2.5 m. However, the sea surface wind direction changed significantly approximately 6 hours before the accident. The gradually enhanced northerly/northwesterly wind forced the growth and development of a wind sea, which later transformed into a swell. The "new" swell therefore had a markedly different direction from that present approximately 6 hours before. The freshly generated swell and wind sea both had smaller $\Delta T$ and $\Delta D$ values, producing favorable conditions for coupling between the swell and the wind sea and leading to possible generation of waves dangerous to ship safety. In the second case, the overall sea state was quite rough, with an $H_s$ higher than 8 m. Although the sea surface wind speed increased gradually before the accident occurred, the sea surface wind direction remained southwesterly. On the other hand, although $\Delta T$ remained quite small (approximately 1 s) for more than 12 hours, $\Delta D$ exhibited significant variation, decreasing from more than 100° at 9 hours before to approximately 60° when the accident occurred. A plausible explanation is that the area was experiencing an extra-tropical cyclone, which had a rotational sea surface wind field and also moved continually. This cyclone therefore generated swells that propagated to multiple directions. Detailed analysis of the sea states associated with these two specific cases further demonstrates that both oblique wave directions and similar wave periods between the wind sea and the swell are two key factors of crossing seas that can lead to the generation of sea state dangerous to shipping safety.

The dynamic wave interactions presented in the two specific cases analyses demonstrated that the oblique wave directions (40° ~ 60°, listed in the cases) and the narrow wave periods between wind sea and swell led to the increase in the wave height, which could be an indicator of wave energy transformation and worsening sea state. This result is consistent with the explanation given in the previous paragraph.

Although the accuracy of the model data were validated using ERA-40 and operational archive results (Poli et al., 2013),

Finally, ship safety could be improved if the major contributors to dangerous sea states are identified and monitored, especially  along major shipping lanes. In future work, the use of multisource data will undoubtedly provide a more complete description of these complex phenomena.

**Acknowledgments**

This study was partially supported by grants from the National Natural Science Foundation of China (No. 41406198), the "Pioneered Hundred Talents Program, Chinese Academy of Sciences" and the Hainan Key S&T Programme (No. ZDKJ2016015). Additionally, we acknowledge that the global ship accident database and the ERA-20C Ocean Wave dataset were accessed from the IMO Global Integrated Shipping Information System (https://gisis.imo.org/Public/Default.aspx/) and the ECMWF Public Datasets web portal (http://apps.ecmwf.int/datasets/data/era20c-wave-daily/type=an/), respectively.

**Responses to reviews**

We would like to thank the reviewers for their evaluation of this study and the helpful suggestions and comments. Comments are responded on a point-by-point basis. Please check the revised manuscript for the details. We hope this revised version could satisfy the reviewers and be acceptable for publication in NHESS.

**Responses to Reviewer 1**
* * *
It is an interesting paper investigating the correlation between ship accidents and certain sea states, co-occurrence of wind sea and swell, in particular. It is well known that ships are endangered in high seas, depending on their size. Therefore, the IMO recommends ship owners to take advantage of ship routing services. However, accidents also occur unexpectedly at moderate sea states and thus it makes sense to dig deeper in the available data base. From this point of view, I find this paper is worth being published. However, I have some major concerns and proposals of improvement:

- The wave model data set ERA-20c used is not suitable for this study, therefore acquire a high resolution data set

- Discuss the typical cases in terms of time series in addition to the 2-dim- representation

- Include interpretation of wave spectral partitioning

- Include some discussion on ship behavior in rough seas

Under these conditions, the paper will be acceptable, otherwise, I would have to reject the paper.

**Response:** Thank the reviewer for the positive comments. We have made a major revision following the reviewers' comments.

- *The wave model data set ERA-20c used is not suitable for this study, therefore acquire a high-resolution data set.*

The detailed answer for this question presented in the point by point answers. Please refer to the response for comment 2.

- *Discuss the typical cases in terms of time series in addition to the 2-dim-*

*representation*

Time series of sea states for the case studies have been added to the revised manuscript. Please refer to Figure 4 and Figure 7 of the revised manuscript.

- *Include interpretation of wave spectral partitioning*

For the first case study, interpretation of wave spectral partitioning has been added to the revised manuscript. Please refer to the last paragraph in Page 10.

- *Include some discussion on ship behavior in rough seas*

We didn't discuss ship behavior in rough seas in details, as the present study focuses on investigating sea states impacts on ship accidents. Some general discussion on this was added to the revised manuscript. Please refer to the first paragraph in Section 3.

**Comment 2:** Numerical Wave Model Data:

Page 3, line 24-25:

The Ocean Wave Daily data in the ERA-20C dataset are available from 1900–2010 every 3 hours at a grid size of 0.125°ERROR. The spatial resolution is 1.5° ……

**Response:** The ERA-20C data with various spatial resolution (from the lowest 3° by 3° to the highest 0.125° by 0.125°) are available. We used the ERA-20C data with the highest resolution of 0.125° by 0.125° for this study. The following figure shows the screenshot of the data downloading interface (at http://apps.ecmwf.int/datasets/data/era20c-wave-daily/type=an/)

But we also noticed that the ERA-20C data with the highest resolution of 0.125° by 0.125° should also have some biases, particularly when the reviewer pointed out that the modeled wave height of the first case (presented in the original manuscript). We had also checked that case again and confirmed this point. The detailed explanation why the modeled wave height of the first case has such distinct bias was presented in the interactive discussion panel. On the other hand, spatial resolution of coastline used in the ERA-20C data is rather low, which can lead to that the model results seem to be very smooth in the offshore area, i.e. they don't represent fine structure of sea state in the offshore areas. We have added some discussions on this point in Section 6.

[Figure]

**Fig.R1** The download interface of ERA-20C data

**Comment 3:** Overview of Ship Accidents

The section describes the large variety of ship types and accidents. It is clear that each case would deserve a distinct study taking into account the ship's properties. Since this will hardly be possible, some discussion on ship's behavior in heavy seas would be helpful at this point. The authors already mention parametric rolling, but a lot other dangerous incidents may occur, like extreme slamming, bending and torsional stresses, green water on deck and emerging propellers (both reducing ship's stability).

**Response:** Thank the reviewer for the valuable suggestion. The initial event in the IMO reports describes the triggering behaviors of each accident. According to initial event, we have divided five types in terms of ship behaviors. In order to facilitate statistics, the classification is not based on a detailed trigger factor but a general result. It is no doubt that the ship behaviors in rough seas mentioned by the reviewer can induce stranding/grounding or foundering/sinking. We have added some discussion on this point according to reviewer's suggestion. Please refer to the first paragraph of Section

3.

**Comment 4:** Analysis of the Sea State during Ship Accidents

(1) "With the statistical evaluation in this section the authors attempt to find evidence of a relation between ship accidents and crossing seas (wind sea and swell), characterized by small $\Delta T<2s$, directional spread of 30 - 40°.

Indeed, a tendency in this direction can be recognized, but there is a lot of noise in the data, which remains unexplained. Some portion of this noise is probably due to the coarse data grid. The rest should be explainable by the large variety of ships and different kinds of causes for accidents (see 3.)."

**Response:** Thank the reviewer for pointing out of the noise data question. With respect to the used ERA-20C data, as we response to Comment 1, it does have a grid size of $0.125° \times 0.125°$. But the coastline used in the dataset has relatively low coarse resolution. As a result, some biases should exist in the offshore areas. On the other hand, even encountering the same sea state, different ships should respond differently. This phenomenon may also lead to a variety of statistical results in a variety of situations. We have added some discussions on this point. Please refer to line 9 -14 in page 16 of the revised manuscript.

(2) "Human errors may also cause accidents, even in moderate sea states. A discussion of this issue should be included here."

**Response:** Thank the reviewer for pointing it out. We missed this point in the previous manuscript. But as stated above that we would like to focus on sea state studies, we just mentioned it in the revised manuscript, while don't expand it for a detailed discussion. Please refer to the sentences in line 12 -15, page 3 of the revised manuscript.

(3) "Wave steepness values between 0.03 and 0.04 are surely not hazardous for ships at all, but steepness of individual waves will probably increase the average steepness. The positive correlation between steepness and the spread between sea and swell propagation is interesting: Is there some theoretical explanation for this phenomenon?"

**Response:** It is a good question. When we found this interesting feature, we also tried

to find some theoretical explanation, but we have not solved it yet. This remains our further study.

**Comment 5:** Sea States of Typical Cases

(1) "It should be mentioned that the distinction between wind sea and swell is an artificial construction. Mariners usually think of swell as a wave system originating from a distant storm, travelling into the local wind field. Crossing seas of this type are not very frequent but very hazardous. The sea states discussed in this section and presumably most of the other 753 cases do not involve a "classical" swell. Crossing seas with angles of 30-40° between directions of wind sea and swell are typically generated by rapidly moving low pressure systems, particularly in the vicinity of cold fronts with sudden changes in wind direction. Consider a storm with high wind sea: If the wind changes direction, a new wind sea is generated and the "old" wind sea is transformed to "swell". This partitioning of wave systems is done by the wave model post processing."

(2) Figures 4 and 5 are difficult to interpret, I could hardly follow the arguments in the text. The coarse 1.5° model grid creates strange rectangular wiggles of contours. Furthermore, I miss date and time in the legend.

(3) "In order to thoroughly investigate the cause of accidents I suggest a local description in terms of time series like the ones I include here, based on the operational ECMWF Global WAM."

**Response: We did a major revision with respect to the case study.**

*First*, As the reviewer pointed out, the modeled wave height of the first case was wrong. We had confirmed this point and explained why the modeled wave height has such distinct difference from the operational WAM results. This has been presented in the interactive discussion panel and is not presented here anymore. Therefore, the original case was replaced with the other one of the 58 swell-related cases. This is presented in Section 5.

*Second*, By coincidence, the new case of crossing sea state of wind sea and swell fits very well with the reviewer's consideration on crossing sea state. The new case does

indicate a "new" swell system transformed from local wind sea and the local wind sea eventually composes of a crossing sea state which is a great threaten to the ship safety.

*Third*, interpretation of spectral partitioning on this case was also added. Please refer to the last paragraph of page 10.

*Fourth*, analysis of the second case was re-written.

*Fifth*, Time series of sea state and wind field of the two cases at the accident locations are presented in Figure 4 and Figure 7, respectively.

*Sixth*, The two-dimensional sea state diagrams were updated to exclude swell period maps for better interpretation. Please refer to Figure 5 and Figure 8 of the revised manuscript.

**Comment 6:** Minor revisions

(1) Page 2, line 5: A high wave height is  undoubtedly a threat 5 for ships, yet some ships wreck at relatively low wave heights  but high wave steepness sea states (Toffoli et al., 2005).

**Response:** It is done as suggested.

(2) Page 2, line 28-29: The detailed information discussed above  is presented in section 2.

**Response:** It is done as suggested.

(3) Page 3, line 22: "ERA-Interim" has not been described so far.

**Response:** "ERA-Interim" have revised to "ERA-Interim dataset (12 h)"

(4) Page 5, line 1: The number of ship accidents  inside each region was summed …

**Response:** It is done as suggested.

(5) Page 6, line 2: As discussed in the Introduction , the co-occurrence of wind sea and swell conditions is considered a potential causal.

**Response:** It is done as suggested.

(6) Page 8, line 4: a small ΔD area in the northerly direction and a large ΔD area in the southerly direction. seen from the ship's position? This is hardly recognizable on the map of fig 4!

**Response:** Please check whether the new Figure 5 yields better visual interpretation.

**Responses to Reviewer 2**
* * *
**Comment 1:** Abstract: -sea state conditions play a significant role in shipping safety. should be sea state. remove the condition.

**Response:** The suggestion has been followed in the revised manuscript.

**Comment 2:** The sea state parameters, including the significant wave height, the mean wave period and the mean wave direction, obtained from numerical wave model data were analyzed for selected ship accidents. This sentence is fuzzy. Please rewrite it.

**Response:** The sentence was revised as: "Sea state parameters of a numerical wave model, i.e., significant wave height, mean wave period and mean wave direction, were analyzed for the selected ship accident cases." Please refer to Page 1, line 13 - 14.

**Comment 3:** Introduction -wave period (T) cross-zero wave period or other types of wave period?

**Response:** Yes. It is cross-zero wave period. It was clarified in the revised manuscript. Please refer to Page 2, line 1.

**Comment 4:** Date and method -the ERA-20C products describe the spatio-temporal evolution of the atmosphere (on 91 vertical levels, between the surface and 0.01 hPa), the land-surface (in 4 soil layers), and ocean waves (for 25 frequencies and 12 directions). I understand you want to describe the high-quality of ECMWF-20C data, however, waves are used in this study. Thus the atmosphere and soil are useless here.

**Response:** Yes. The sentence is a little bit lengthy and we have revised as: "The ERA-20C products describe the spatio-temporal evolution of ocean waves for 25 frequencies and 12 directions." Please refer to Page 3, line 23 - 24.

**Comment 5:** I wonder why not analysis the relation between ship accident and winds?

especially in poor weather, wave should be related with wind.

**Response:** It is a good question. Sea wind does have significant impact on shipping safety and in many cases the high waves induced by wind can cause serious ship casualties. However, in this study, we would like to know impacts of sea state when both windsea and swell present on shipping safety. Swell are long waves propagating far away from generation sources and therefore are not effected by sea wind anymore. Therefore, in this study, we didn't investigate relation between sea wind and ship accidents. We have clarified this point in the revised manuscript. Please refer to Page 6, line 6 - 10.

**Comment 6:** 4.1 Wave Height Here, the variable is not coincident with description above. Following the manuscript, SWH is right!

**Response:** The suggestion has been followed. Please refer to Section 4.1. of the revised manuscript.

**Comment 7:** Figure 4 the figure at forth row should be replotted due to the colors does not overlap with x axis and y axis

**Response:** Thank the reviewer for pointing the slight offset of the two plots. We have modified the offset in the new figure. Since the figure 4 in the old manuscript has been replaced by Figure 5 of the revised version. Please refer to the third row of Figure 5 in Page 11.

**Comment 8:** Figure 5 the arrow is out of area

**Response:** Starting points of the arrows are the grids where the model data are available. The boundary (axis limits) of the plots are the same as the grids, therefore, it is unavoidable that some arrows are beyond the plots. In the revised manuscript, there are three figures that contain the arrow graph (i.e. Figure 5, 6 and 8). Then the situation of arrow out of area still exist in the three figures.